# Cancer Stem Cells—The Insight into Non-Coding RNAs

**DOI:** 10.3390/cells11223699

**Published:** 2022-11-21

**Authors:** Rut Bryl, Oliwia Piwocka, Emilia Kawka, Paul Mozdziak, Bartosz Kempisty, Agnieszka Knopik-Skrocka

**Affiliations:** 1Section of Regenerative Medicine and Cancer Research, Natural Sciences Club, Faculty of Biology, Adam Mickiewicz University, 61-614 Poznań, Poland; 2Department of Electroradiology, Poznan University of Medical Sciences, 61-701 Poznań, Poland; 3Doctoral School, Poznan University of Medical Sciences, 61-701 Poznań, Poland; 4Prestage Department of Poultry Science, North Carolina State University, Raleigh, NC 27695, USA; 5Department of Human Morphology and Embryology, Division of Anatomy, Medical University of Wrocław, 50-367 Wroclaw, Poland; 6Department of Veterinary Surgery, Institute of Veterinary Medicine, Nicolaus Copernicus University, 87-100 Torun, Poland; 7Department of Cell Biology, Faculty of Biology, Adam Mickiewicz University, 61-614 Poznań, Poland

**Keywords:** cancer stem cells, non-coding RNAs, EMT, phenotypic and metabolic plasticity, anti-cancer therapies

## Abstract

Since their initial identification three decades ago, there has been extensive research regarding cancer stem cells (CSCs). It is important to consider the biology of cancer stem cells with a particular focus on their phenotypic and metabolic plasticity, the most important signaling pathways, and non-coding RNAs (ncRNAs) regulating these cellular entities. Furthermore, the current status of therapeutic approaches against CSCs is an important consideration regarding employing the technology to improve human health. Cancer stem cells have claimed to be one of the most important group of cells for the development of several common cancers as they dictate features, such as resistance to radio- and chemotherapy, metastasis, and secondary tumor formation. Therapies which could target these cells may develop into an effective strategy for tumor eradication and a hope for patients for whom this disease remains uncurable.

## 1. Introduction

Malignant tumors are one of the most prevalent causes of death world-wide. According to Globocan (https://gco.iarc.fr/today/home (accessed on 16 December 2021)) in 2020, the estimated number of new cases reached almost 20 million, while the number of deaths was close to 10 million. Breast, colon, lung, and ovarian cancers are the most often diagnosed tumor type in women over 40 years of age. The highest morbidity and mortality in men is observed for lung, prostate, and colon cancers. Therefore, many scientists have been focused on these solid tumors with regard to disease etiology and treatment. It was found that cancer cells are heterogenic, and that only a small fraction are responsible for tumor development and metastasis. It was demonstrated that these cells are able to initiate the tumor growth when implanted in mouse hosts. Hence, they are called tumor-initiating cells or cancer stem cells (CSCs) [1].

The history of CSCs ability to drive tumorigenesis is long, but the real breakthrough came in the 1990s and 2000s [2,3,4,5,6]. The low number of CSCs, occurring at a frequency of 1–100 in 10^−6^ [2] is a serious obstacle for cell isolation and similarly, and non-specific markers of CSCs of different origin remain challenging for cell identification [7]. However, it was found that CSCs show many specific features, such as stemness (proliferation capacity and self-renewal), phenotypic plasticity, metabolic reprogramming, and drug resistance [8,9]. With regards to phenotypic plasticity, epithelial-mesenchymal transition (EMT) activation is linked to the formation of CSCs [1]. However, the underlying mechanism is still unclear. CSCs are also known to perform in immune evasion from immunotherapy. The recruitment of M2 macrophages, Tregs, dendritic cells [10], and involvement of non-coding RNAs may perform a role in suppressing tumor development [11].

In recent years, intensive studies on epigenetic changes in malignant tumors have been conducted, including the role performed by non-coding RNAs [12,13,14]. Understanding the regulation of gene expression by miRNAs (microRNAs), lncRNAs (long non-coding RNAs), and circRNAs (circular RNAs) has been of particular interest. These non-coding RNAs can play both oncogenic and suppressor roles in cancer and exhibit tumor-specific expression [15,16,17]. The strong influence of CSCs microenvironment on their biology, including therapy resistance, is an important topic for scientific research [18].

It is also important to consider the state-of-art in CSC biology, especially in the context of their remarkable plasticity and some aspects of underlying epigenetic mechanisms and potential targets for future therapeutic strategies against CSCs.

## 2. Biology of CSCs and Their Interaction with Tumor Microenvironment

CSCs show complex biology. It is the effect of specific features and crosstalk with tumor microenvironment [19,20,21,22,23,24]. Key features of CSCs, including stemness, self-renewal, phenotypic changes strongly connected with EMT phenomenon, metabolic reprogramming, and ability to become invisible to immunological system are presented in Figure 1. CSCs plasticity, phenotypic and metabolic, is their ability to dynamically change under microenvironment conditions. It is the result of different mechanisms regulated by both cell intrinsic and extrinsic factors [25]. These signals influence the expression of various genes in CSCs (Table 1). BMI1 and Sox2 can regulate plasticity and pluripotency [26,27]. The EMT can be induced by SOX9 [28]. Tumor hypoxia triggers metabolic reprogramming and phenotypic plasticity [29]. Oxygen insufficiency causes dimerization of HIF-1α and HIF-1β to form the HIF-1 complex [30]. Silencing of HIF-1α can suppress the expression of stem cell genes, in particular OCT4, SOX2, NANOG, and KLF4, thus preventing the progression of cancer [31].

Hypoxia in the tumor microenvironment is common in advanced cancer and is related to poor prognosis and a worse survival rate. A growing body of evidence has discovered that hypoxia can promote cancer cell invasion, metastasis, and EMT. All these events can promote stemlike characteristics in cancer cells. Hypoxia-inducible factor 1 (HIF1) is a vital molecule in the regulation of CSCs. HIF1 is involved in tumor growth, immune evasion, and metabolic reprogramming. Thus, HIF1 appears to play an essential, if not critical, role in the formation and preservation of CSCs [31].

### 2.1. CSCs Signaling Pathways

There are few signaling pathways activated in CSCs. The most commonly disturbed signaling cascades in CSCs are phosphatidylinositol 3-kinase/Akt/mammalian target of rapamycin (PI3/Akt/mTOR) pathway, Janus kinase/signal transducers and activators of transcription (JAK/STAT) pathway, and Wnt and Notch [45].

#### 2.1.1. PI3K/Akt/mTOR Signaling Pathway

PI3K signaling pathway is aberrantly activated in a variety of cancers, and thus performs a vital role in tumor growth and proliferation [46]. Abnormal functioning of the PI3K/Akt/mTOR pathway was found in ovarian [47], breast [48], and prostate cancers [49]. In glioblastoma, the PI3K/Akt/mTOR pathway is activated by loss of the *PTEN* gene, which leads to enhancement of tumor cell progression by Akt [50]. A signaling molecule like PTEN is known as an important suppressor of tumor growth [51,52]. Hence, the mutations in *PTEN* often promote uncontrolled cell growth, resistance to apoptosis, and higher migration. Activation of mTOR promotes the proliferation of breast CSCs and nasopharyngeal carcinoma stem cells [9]. Oncogenic induction of PI3Kα is related to initiation of EMT, which is identified with increased plasticity [53] (Figure 2).

#### 2.1.2. Wnt Signaling Pathway

The Wnt signaling pathway consists of three distinct cascades: (1) the canonical Wnt pathway (involving β-catenin and T cell-specific transcription factor (TCF), and lymphoid-enhancer-binding factor (LEF)); (2) the non-canonical pathway which is β-catenin independent; and (3) the non-canonical Wnt-calcium pathway, which regulates intracellular calcium levels [54]. The canonical Wnt pathway is associated with the growth and self-renewal of stem cells. It takes place in the proliferation and differentiation of progenitor cells. At transcriptional and posttranscriptional levels, it controls cell fate [45,55]. The non-canonical pathway is involved in the control of the cytoskeleton. It affects the planar- cell polarity and is responsible for inhibition of the canonical signaling [54,55]. Wnt signaling participates in cancer initiation by generating CSCs from normal stem cells or by re-acquisition of stemness in subpopulations of cancer cells [56].

The Wnt signaling pathways are crucial for the development of CSCs, but their crosstalks with other cascades (FGF, Notch, Hedgehog, and TGFβ/BMP) are also essential for the regulation of CSCs markers expression [55]. For example, deregulation of the Wnt pathway in CSCs correlates with increased CD44, *MMP7, HAS2, CXCR4, CLDN1,* and *FN1* expression, which implicates the possibility of metastasis and chemoresistance. Genes, such as *LGR5* and *DCLK1,* are also targets of Wnt and are responsible for tumor initiation in CSCs [56]. A particularly important target of Wnt is the *ABCB1* gene, which is an ATP-dependent molecule transporter. Overexpression of *ABCB1* and its product P-glycoprotein is associated with multiple drug resistance (MDR), which is directly related to chemo-resistance. The promoter of *ABCB1* is composed of several β-catenin/TCF4/LEF1-binding sites, which implies regulation of *ABCB1* through the canonical Wnt/β-catenin pathway in colorectal and breast cancers [56].

#### 2.1.3. JAK/STAT Signaling Pathway

The JAK/STAT pathway regulates stem cell development, hematopoiesis, and inflammatory response. It carries a signal from cytokines, interleukins, and growth factors that impact transmembrane receptors, such as the family of epidermal growth factor receptors (EGFR) [57]. The binding of a ligand induces a conformational change in the receptor what causes the positioning of receptor-associated JAKs. This arrangement facilitates phosphorylation of appropriate tyrosine residues that modifies inactive JAKs into catalytically active tyrosine kinase. Activated JAKs tyrosine residues in the cytoplasmic region of the receptor, enable the formation of special binding sites where signal transducers and activators of transcription (STATs) may attach. The STATs are set up in dimers and proceed to the nucleus where can bind to interferon-stimulated response element (ISRE) which modify the transcription of genes regulating proliferation, differentiation, and apoptosis of the cells [57,58]. Pieces of evidence state that the JAK/STAT pathway is commonly activated in CSCs have been found in stem-like cells derived from breast, colon, prostate, ovary, and lung cancer [59,60,61]. The JAK/STAT pathway stimulates not only the breast cancer progression and inflammation, but also increases the conversion of non-stem cancer cells into breast CSCs [59]. Moreover, there is a correlation between the activation of the JAK/STAT pathway and stemness features, stating that the silencing of JAK/STAT signaling decreases the self-renewal of colorectal cancer stem cells [61] (Figure 3).

#### 2.1.4. Notch Signaling Pathway

The family of Notch receptors comprises of four members (Notch 1–4) which can bind five various ligands, such as Delta-like ligands 1, 3, and 4, and Jagged ligand 1 and 2 [62]. The Notch signaling is a conserved pathway regulating cancer cell conservation, nourishment, and general support of cancer growth. *NOTCH* can be upregulated and downregulated. Upregulation may occur in the brain and pancreatic tumors (Notch 1, 2, and 3), breast cancer (Notch 1, 2, 3, and 4), and downregulation is typical for colorectal cancer (Notch 1, 2, and 3). In breast cancer, the functioning of *NOTCH* genes is diverse, corresponding to the molecular subtype of tumor, i.e., *NOTCH1* can induce *HER2* transcription and cause an increase in the mammary stem cells and breast CSCs [63]. Additionally, overexpression of *NOTCH1* gives the worst prognosis and survival outcome. Studies on the MCF-10A cell line have shown that overexpression of *NOTCH1* causes cell transformation, change in cell shape, increase in cell growth, and acquisition of resistance to apoptosis [64]. Since the Notch pathway serves for stem cell differentiation, it can promote the development of ER+ luminal tumors. The upregulation of Notch2 and Notch3 participates in the differentiation of progenitor cells to the ones of luminal type, and Notch3 increases invasion [63,65]. On the other hand, Notch4 is prevalent in the breast CSCs population and is present among basal cells [66]. In an ovarian tumor, expression of the high level of the *NOTCH3* corresponds to increased drug resistance and poor overall survival [67]. Generally, the most significant cancer stem cell markers related to the Notch pathway are CD133, Musashi-1, CD44, EpCAM, CD166, and Bmi1 [63].

Notch signaling activation contributes to the appearance of tumorigenic conditions, corresponding to the type of tissue, receptor-ligand interactions, and genetic mutability [63]. Moreover, Notch signaling plays a significant role in the regulation of asymmetric division and cell stemness [41]. Increasing evidence indicates the importance of the Notch pathway in the linkage between angiogenesis and CSCs, which underlies the validity of targeting CSCs in terms of therapy [68].

Notch signaling promotes activation of genes required for epithelial-to-mesenchymal transition, which is one of the most important processes described in CSCs. The EMT is an intrinsic event for the metastatic spread of the tumor. Evidence suggests, activation of Notch1 causes suppression of E-cadherin, which leads to EMT occurrence in breast cancer cells. Additionally, activation of Notch by hypoxia conditions leads to downregulation of E-cadherin and β-catenin, thus increased cell migration and invasion of breast cancer cells cultured in low-oxygen conditions is observed [64] (Figure 4).

### 2.2. CSCs Plasticity, EMT and Dormancy

Phenotypic plasticity of CSCs can be found as ability to switch states (phenotypes) as a response to tumor microenvironment conditions [69]. A strong relationship exists between phenotypic plasticity, stemness, and EMT [24,25]. During EMT, cells show changes in their morphology and expression of genes towards a stem-like state [25]. EMT and ECM remodeling have a strong impact during tumor angiogenesis. CSCs are able to form vessel-like structures, which were described as vasculogenic mimicry (VM) [70]. The mechanism of VM is not dependent on endothelial cell proliferation and VEGF. It is strongly connected with phenotypic changes [71]. CSCs transdifferentiate to cells with some features of endothelial cells. It was demonstrated that cancer cells involved in VM show expression of CD133, CD44, ALDH, and Sox2 [72,73], but also VE-cadherin and CD31 [19,74]. The meta-analysis of thirty-six studies has revealed that VM is associated with shorter overall survival and is a poor prognostic factor [75].

The EMT process corresponds to a wide spectrum of biological variances triggering the conversion of cells from epithelial to the mesenchymal state. Cells that undergo EMT, acquire migratory and invasive properties [76]. Epithelial cells exhibit more proliferative features, while mesenchymal cells have an increased tendency for migration and ability to impact stroma through matrix metalloproteinases (MMPs) [40,77]. Moreover, epithelial cells show overexpression of markers, such as E-cadherin or members of miR-200 family, while for the characterization of mesenchymal cells markers, such as N-cadherin, vimentin, or fibronectin, can be used [76]. The signaling pathways like Wnt, Notch, Hedgehog, or Myc act as EMT inducers and are related to stemness properties of CSCs, and impact migration and invasion [78]. Extracellular EMT inducers comprise TGF-β, EGF, Axl-Gas6 pathway, hypoxia, and ECM elements. In addition, transcription factors (TFs) Twist1, Snail1, and Zeb1/2, T-box TF Brachyury promote EMT [79].

EMT was first observed by Greenburg and Hay in 1982 [80] and named epithelial-mesenchymal transformation. EMT phenomenon is crucial for normal embryonic development, but also is linked to several pathological processes, including wound healing, fibrosis, and cancer progression [81]. In the case of tumor cells which can express typical markers for both cell states (epithelial and mesenchymal), the term transformation was replaced with transition [82]. Decades of studies have revealed that cancer cells are able to form not only fully epithelial or fully mesenchymal cancer cells, but also various hybrid E/M (intermediate states). In 2020, the EMT International Association (TEMTIA) published a work with the current status of the knowledge about EMT and nomenclature [83]. According to this data, EMT should be treated as the ability to progress along the epithelial–mesenchymal axis and to adopt different intermediate hybrid E/M states [82,83]. The hybrid phenotype is critical for the maintenance of tumorigenicity of basal breast cancer cells. Highly tumorigenic cell population with expression of CD104/CD44 cell surface antigen and transcription factors Zeb1 and Snail1 was isolated in a hybrid E/M state. For this cell population increased expression of Snail and Wnt signaling pathway was also observed [84].

Phenotypic plasticity of cancer cells and their ability to undergo EMT and metastasis can be modulated by tumor microenvironment. There are many extracellular factors determining the plasticity of CSCs [85]. It is the result of crosstalk leading by CSCs and cellular components of tumor microenvironment, like cancer-associated fibroblasts (CAFs) and macrophages.

CAFs can modulate CSCs plasticity via different signaling pathways; for instance, in lung cancer by IGF-II/IGF1R signaling pathway [86] and in hepatocellular carcinoma through c-Met/FRA1/HEY1 signaling [87]. In prostate cancer, CXCL12 expressed by CAFs interacts with CXCR4 on tumor cells, induces EMT, and promotes metastasis [88]. The crosstalk between CAFs and cancer cells promoting their phenotype changes and metastasis was also observed in breast cancer [89]. The enhancement of stem cell features appeared by the activation of the Notch mechanism. Macrophages can secrete factor Oncostatin-M, an IL-6 family cytokine, which activate the dedifferentiation of triple negative breast cancer cells into aggressive stem cells [90]. With regards to the crosstalk between cancer cells and macrophages in a tumor, studies in silico co-culture models have revealed that macrophages (M1 and M2 phenotypes) can alter the epithelial vs. mesenchymal state of cancer cells. These results may be helpful for efficient therapeutic strategies.

EMT is strongly implicated in tumor relapse. According to the studies of Sun et al. conducted on prostate cancer, EMT can be induced by androgen deprivation, which is the first-line therapy [91]. It is probable that the feedback loop involving the androgen receptor and the Zeb1 transcription factor are responsible for the transition. The castration-resistant prostate cancer is a major clinical problem, and these results seem to be crucial in terms of medical implications for second-line treatment of castration-resistant prostate cancer. The most recent studies of Guo et al. showed that Numb protein performs an important role in xenograft prostate tumor growth and castration-resistant prostate cancer as a suppressor of CSCs Notch and Hedgehog signaling [92]. The inhibition of the Notch and Hedgehog signaling pathways significantly increases apoptosis in Numb^−/low^ cells in response to androgen-deprivation therapy.

CSCs show their plasticity not only in ability to EMT. These cells can use adaptive and protective mechanism known as dormancy [93]. In this state cancer cells stop proliferating. Clinically, cancer dormancy is very difficult to detect, and it is defined as remission time. Two categories of dormancy in cancer can be distinguished. First, cellular dormancy, means that each cancer cell shows cell cycle arrest. The second is tumor dormancy when in cancer a balance between growth and apoptosis rates appears. The length of dormancy is long in prostate cancer or in hormone dependent breast cancer, whereas in triple negative breast cancer this period is shorter [94]. Moreover, it is suggested that dormancy of CSCs can be the result of epigenetic changes in CSCs. The nature of dormancy is reversible, and the epigenetic mechanisms can be responsible for regulating, maintenance, and reactivation of cancer cells from the dormancy [94]. Non-coding RNAs seem to be very important regulators of dormancy in cancer, e.g., the angiogenesis/dormancy switch [93]. In addition to non-coding RNAs, there are several other intracellular and extracellular signals involved in the mechanisms of dormancy and reactivation, to the group of dormancy signals belong, e.g., p16, p21, p53, TGFβ2, or BMP4, and for reactivation Pi3K/AKT, TGFβ3, and HIF-1α can be responsible [95].

Some of anticancer drugs, like fluorouracil, increase the number of dormant cancer cells and enrich the population of CSCs, which leads to chemotherapy resistance [96]. However, some of chemotherapeutic agents show the opposite action. They can reactivate dormant cancer cells. The studies of Gao et al. on head and neck squamous cell carcinoma have revealed that LB1, an inhibitor of protein phosphatase 2, can enhance the cytotoxic sensitivity to chemotherapy via promoting entering of cancer cells from dormancy into the cell cycle [95].

### 2.3. CSCs Metabolic Changes

Metabolic plasticity, as another adaptation to microenvironmental conditions, is one of the most important hallmarks of cancer cells [97]. Cells can modify metabolism using different energy sources to enhance their survival and maintain homeostasis. With regards to the heterogeneity of tumor cells, CSCs show higher metabolic plasticity compared to normal cancer cells [20]. In non-stem, highly proliferative cancer cells, glycolytic phenotype is observed with high glucose uptake, low oxygen consumption, low mitochondrial mass, and ROS. These features are typical for the Warburg effect where the predominant pathway for ATP generation is glycolysis [51]. In glucose-deprived conditions, CSCs tend to shift into a quiescent (non-proliferative) state and depend on OXPHOS to produce ATP. Quiescent CSCs show oxidative phenotype and reverse Warburg effect metabolism with high oxygen consumption. Moreover, CSCs with reduced proliferation are more resistant to chemotherapy, which targets mostly proliferative CSCs [98]. The changes in metabolism can be observed not only in cancer cells, but also in cells of tumor microenvironment, including upregulation in lactate production and the acidification of the tumor’s stroma [99].

Switching between glycolytic and OXPHOS phenotype in CSCs is suggested. These cells can exist in a hybrid metabolic state [100]. Hence, both glycolysis and OXPHOS should be blocked. Metformin, as a glycolysis reducer and OXPHOS inhibitor cane inhibit this metabolic plasticity [101].

### 2.4. Epigenetic Regulation of CSCs—The Role of microRNAs, circRNAs and lncRNAs

Epigenetic mechanisms regulate potentially heritable changes in gene expression, not associated with changes in the DNA sequence. Such changes dictate cancer formation and progression, leaving their mark on cancer stem cells. There are three main mechanisms of such gene expression regulation, namely DNA methylation, chromatin modification, and non-coding RNAs [102,103]. In this review, we focused on the third mechanism, non-coding RNAs. Non-coding RNAs serve regulatory roles in self-renewal, metabolic plasticity, resistance to radio- and chemotherapy, interactions within the tumor microenvironment or formation of secondary disease foci (Figure 5). They have shown a lot of promise in development of targeted therapies to combat cancer [104,105,106,107]. The use of non-coding RNAs may enable targeting cancer stem cells, and this may become an effective strategy to eradicate cancer.

#### 2.4.1. miRNAs in CSCs

miRNAs are a class of non-coding, endogenous, single-stranded RNAs, approx. 22 nucleotides in length [108], which have been demonstrated to regulate more than 60% of all the protein-coding genes in mammals [109]. The size of human miRNome corresponds to nearly 1% of all genes which undergo expression in human, making it one of the most abundant classes of genetic regulators [110,111].

miRNAs can regulate gene expression on posttranscriptional level by specific recognition and interaction with mRNA. Most often, miRNA–mRNA interactions occur via seed region—a short sequence on the 5′ end of miRNA. Due to this, single miRNA can act on multiple mRNAs and a single mRNA can be regulated by multiple miRNAs [112,113]. The process of gene silencing can proceed in two different manners, depending on the degree of complementarity between miRNA and mRNA molecules. Most commonly, miRNAs interact with the 3′ untranslated region (3′ UTR) of target mRNAs leading to translational repression and mRNA deadenylation and decapping. In cases where there is perfect complementarity between miRNA and mRNA, transcript degradation occurs [114].

Various roles of these small regulatory RNAs in cancer have been characterized. Upregulation or downregulation of the RNA molecules which function as oncogenes and tumor suppressors, have been implicated in tumor development and progression [17]. Recent evidence suggests that miRNAs control important CSCs’ features, such as self-renewal ability, chemo- and radioresistance, metabolic plasticity, or tumor initiation capacity [115,116]. MicroRNAs are involved in crosstalk with tumor microenvironment, and they can be transported via CSCs-derived exosomes and impact gene expression [117], and they may function both as promotors and suppressors (Table 2).

##### miRNAs as Promoters of CSCs

miR-31 is a promoter of mammary stem cell maintenance and breast cancer development and progression. miR-31 promotes growth, proliferation, and development of malignant features of tumors from MMTV-PyVT mouse model for breast cancer. miR-31 was enriched in breast cancer stem cells (BCSCs) (CD24^+^CD90^+^) and an around 5-fold increase in comparison to CD24^−^CD90^−^ was observed. BCSCs expansion and tumor-initiating ability are favored by this miRNA. Loss of miR-31 marks impact on metastasis as evident by reduced levels of EMT-associated factors and decreased levels of metastasis inhibitors, gain of mesenchymal-like phenotype, and, importantly, reduced lung metastasis and higher metastasis-free survival rate in PyVT/KO mice. miR-31 induction is orchestrated by NF-κβ pathway. miR-31 itself directly targets the mRNAs of *AXIN1, GSK3Β,* and *DKK1*, thereby activating Wnt-β-catenin pathway. Furthermore, it was revealed that TGF-β signaling was elevated in miR-31 KO mammary glands, suggesting miR-31 involvement in repression of this pathway [118].

miR-128-3p is a potential promoter of lung cancer stem cells (LCSCs). This miRNA exhibits high expression in chemoresistant, metastatic A549-luc-CDDP-4th cells in NSCLC cell lines (7-fold as compared to A549-luc-Ctrl-4th cells), a metastatic murine lung cancer cell line LL/2-luc-M38 (70-fold compared to primary normal lung epithelial cells (NLE), and immortalized human bronchial epithelial cell line (BEAS-2B)), and tumors are resistant to CDDP-3rd and CDDP-4th treatments (5- and 60-fold as compared to Ctrl-3rd and Ctrl-4th treatments). Curiously, it has been revealed that only continuous treatment with CDDP at a high concentration (7 mg/mL) for 7 days induced miR-128-3p expression. miR-128-3p was associated with NSCLC clinical stage and TNM stage. As compared to low-expression group, patients with high miR-128-3p levels had shorter median overall survival and progression-free survival. It has been also demonstrated that expression of this molecule is an independent prognostic factor for non-small cell lung cancer patients. miR-128-3p seems to perform crucial roles in promoting EMT, CSCs’ programming, and therapy resistance in NSCLC. miR-128 influences cell morphology, upregulates EMT markers, and increases migration and invasive capacities. Overexpression of miR-128-3p increased levels of stemness markers and favored NSCLC cells’ self-renewal ability. Turning to resistance to chemotherapy, cells with ectopic expression of miR-128-3p demonstrated higher IC50 values for cisplatin, gemcitabine, and paclitaxel. Moreover, this miRNA affects tumor cell proliferation and survival. In vivo, this small non-coding RNA influences cancerogenesis and metastasis in several distal organs. Administration of miRNA sponge and antagomir confirmed that miR-128-3p is an attractive therapeutic target. miRNA exhibits its multifaced activity by targeting 3′ UTR of *AXIN1, SFRP2, WIF1, SMURF2,* and *PP1c* mRNAs, which are negative regulators of β-catenin pathway (first 3) or TGF-β signaling (last 2). This points towards the significance of Wnt/β-catenin and TGF-β signaling in functions played by miR-128-3p in the studied cancer [119].

##### miRNAs as Suppressors of CSCs

miR-145-SNAI1 is a regulatory axis important for colorectal cancer stemness. *SNAI1*, upregulated in colorectal cancer specimens (1.3-fold and 4.5-fold increase), favors stem cell-like phenotype maintenance, while miR-145 acts as a CSCs’ suppressor. It has been shown that this miRNA is downregulated in colorectal cancer cell lines where endogenous SNAI1 was high (SW620: 800-fold and 15-fold downregulation when compared to HCT116 and SW40, respectively) and in SNAI1-overexpressing cell lines (DLD1—36% as compared to vector control). A direct relationship between these two was reported, as Snai1 targets miR-145 promoter and represses its transcription, while overexpression of miR-145 downregulated this transcription factor. miR-145 was also shown to target important stem cell factors such as KLF4, c-Myc, Nanog, or OCT4. The introduction of this miRNA leads to increased radiation and oxaliplatin sensitivity and inhibition of self-renewal. In EpCAM^+^ALDH^+^ cells derived from PDX, *SNAI1* was upregulated, while miR-145 was downregulated which confirms the functionality of the axis in CSCs in vivo [120].

SIRT1-miR-1185-1-CD24 axis is another of such examples. By targeting 3′UTR of CD24 (CRC CSCs marker), miR-1185-1 negatively regulates CSC features and carcinogenesis, which was evidenced by reduced CSC frequency, colony formation, and migration ability in vitro and decreased tumor size and weight in xenograft model. SIRT1 repressed miR-1185-1 transcription by acting as a histone deacetylase, which thereby provides a mechanism for CD24 upregulation in CRC. Furthermore, *SIRT1* and *CD24* exhibited elevated, and miR-1185-1 declined expression in patients with colorectal cancer (5-10-fold downregulation in comparison to healthy colorectal tissue) [121].

miR-141 belongs to miR-200 family, which role in EMT and metastasis was one of the first to be reported. In the work brought here, the authors investigated miR-141 function in prostate cancer stem cells. They revealed that this miRNA is downregulated in CD44^+^ cells derived from both patients and xenografts (10–80% of expression of the corresponding marker-negative populations). Overexpression of miR-141 diminishes cancer stem cell features, including self-renewal. In vivo, miR-141 inhibits tumor formation and regeneration and growth, which is directly linked with negative influence on cell proliferation. Furthermore, miR-141 inhibits metastasis and invasion as shown both in vitro and in vivo. Ectopic expression of miR-141 promotes epithelial phenotype but with a limited depletion of the mesenchymal features. miR-141 targeted *CD44*, members of Rho GTPase signaling pathway (*RAC1, CDC42, CDC42EP3* and *ARPC5*) and *EZH2* mRNAs implying possible mechanisms of its influence on invasion, cell motility, and on cancer stem cells properties [122].

miR-338-5-p and miR-421 have been shown to be downregulated in SPINK1^+^ subtype of prostate cancer (2-fold and 1-4-fold as compared to ERG^+^ subtype, respectively) and negatively regulate proliferation, invasion, and colony and foci formation ability in SPINK1^+^ cell line. In vivo, overexpression of these two miRNAs leads to depletion in the number of intravasated cells and formation of distant metastases, reduced tumor growth, and proliferation. miR-338-5p and miR-421 also appear to be involved in epithelial-to-mesenchymal transition. Importantly, these two miRNAs negatively regulate expression of pluripotency markers, stemness factors, prostatosphere formation and sensitize 22RV1 cells to doxorubicin. In regards to the mechanism of action, these 2 miRNAs modulate *SPINK1* expression by targeting 3′UTR of its transcript and their transcription is, in turn, epigenetically silenced by Ezh2 and its partners in SPINK1^+^ prostate cancer subtype. Interestingly, authors have also provided evidence that the use of epigenetic drugs, including DNMT, HDAC, and HKMT, inhibitors may diminish SPINK1-related oncogenic effects in vitro. The diagnostic potential of evaluating miR-338-5p expression have been shown to be associated with prolonged survival and lower tumor stage [123].

miR-34 is a p-53-regulated tumor suppressor which also negatively impacts CSC function in several cancers, including lung, breast, colorectal, and prostate, among others [125,126,127,128,129,130]. Interestingly, its mimic has been recently tested in clinical trials.

In the study presented here it has been demonstrated that the deletion of miR-34a/b/c modifies intestinal architecture in Apc^Min/+^ mice, leads to enhanced tumorigenesis, proliferation, and reduced apoptosis. These miRNAs possibly affect immune cells in the tumor niche as implied by reduced T-, B-cells, and macrophages frequencies, and observed bacterial infiltration. Cells derived from miR-34a/b/c-deficient adenomas formed intestinal tumor organoids at an increased rate. miR-34a/b/c deletion led to upregulation of mRNAs with sequences matching the miR-34 seed sequence and were associated with EMT, stemness and Wnt signaling, among other cancer-related biological processes. Clinically, miR-34 targets showed upregulation in primary colorectal tumors (1–2 fold as compared to healthy colorectal tissue) and their expression was correlated with lymph-node metastases (*INHBB, AXL, FGFR1*, and *PDFGRB*) and survival (*INHBB* and *AXL*) [124].

#### 2.4.2. circRNAs in CSCs

Circular RNAs (circRNAs) are a class of regulatory RNAs with a covalently closed structure [131]. Studies have revealed high abundance of some circular transcripts, cell- and tissue-type specific expression, evolutionary conservation, and exceptional stability [132,133,134]. The majority of detected circRNAs are exonic circRNAs derived from protein-coding genes and the most frequent mechanism of their generation is termed backsplicing [135,136,137].

Several mechanisms of circRNAs action in cells have been described. The proposed biological roles have been studied only in a small fraction of all identified circular transcripts. Majority of cases involve gene expression regulation by binding miRNAs due to presence of miRNA-binding sites, the so-called miRNA sponging [133,138,139,140]. Beyond this, circRNAs may include RBP-binding sites and by this modulate activity of various proteins, serve as scaffolds for specific enzyme-substrate complexes or recruit proteins to specific cellular locations [135,141,142]. Despite the fact that circRNAs are generally regarded as lacking coding potential, recent evidence suggests that these molecules undergo cap-independent translation [143,144,145]. circRNAs are not as extensively studied as miRNAs, but they have also been proposed as performing in tumorigenesis [16,146,147,148]. circRNAs serve important roles in regulation of CSCs and their major features such as self-renewal, chemoresistance, and pluripotent state maintenance (Table 3).

##### circRNAs as Promoters of CSCs

hsa_circ_002178 can be an example of circRNA, which is implicated in promotion of BCSCs’ features. Li et al. have reported that the hsa_circ_002178 transcript is upregulated in breast cancer tissues (2-16-fold as compared to healthy tissue), and performs a role in stimulation of sphere formation, leading to increased expression of stem cell markers. Furthermore, hsa_circ_002178 overexpression facilitates survival, migration, and invasive abilities. In vivo, knockdown of this circular RNA results in inhibition of tumor growth, and metastasis. hsa_circ_002178 exerts its function by binding miR-1258 to upregulate KDM7A. The expression of circRNA circ_002178 correlates with the survival rate, tumor size, lymph node metastasis, and TNM stage of tumor patients [149].

circAGFG1 is a circular transcript, which activates Wnt/β-catenin signaling in colorectal cancer stem cells. The mechanism underlying Wnt/β-catenin signaling relies on sponging of miR-4262/miR-185-5p, which leads to YY1 upregulation. YY1 activates *CTNNB1* transcription, and subsequently the Wnt/β-catenin pathway. It has been reported that this circRNA is upregulated in colon cancer cell lines and tissues, especially with liver metastasis (6-14-fold as compared to human colon mucosal epithelial cell line; 2.4-fold relative to paired para-tumor tissues; and 1.3-fold as compared to non-metastatic cancer tissues). circAGFG1 favors proliferation, viability, migration, and invasion of colon cancer cells, but influences stem cell characteristics as evident by increased sphere formation capacity, maintenance of CD133^+^ cell population, and upregulated protein levels of CD44, Sox2, Oct4 and Nanog [150].

Lastly, circ_001680 has been found to promote several cancer stem cell characteristics, such as sphere formation ability, elevated expression of stem cell characteristic markers (SOX2, CD44, and CD133), and irinotecan resistance. It also positively influences maintenance of CD44^+^/CD133^+^ cell population, facilitates proliferation, and migration. Circ_001680 was upregulated in colorectal cancer tissues (>1.5-fold as compared to matched adjacent normal tissues), and its expression correlated positively with colorectal cancer tumor stage [151].

##### circRNAs as Suppressors of CSCs

circRGPD6 appears an interesting example of breast cancer stem cells’ suppressor function. Expression of the circRGPD6 circular transcript is significantly lower in BCSCs than in BCCs (1.5-fold), and lower in metastatic than in primary tumors (3.7-fold). circRGPD6 overexpression in vitro contributes to depletion of tumor-initiating properties and pluripotency capabilities of BCSCs as evident by reduced proportion of CD44^+^CD24^−^ cells, suppressed expression of stem cell marker CD44, enhanced expression of DNA damage marker p-H2AX, differentiation markers-Muc1 and α-SMA, and vimentin and E-cadherin. Furthermore, circRGPD6 inhibits self-renewal ability and cell viability. In vivo, circRGPD6 has been shown to inhibit metastasis and proliferation of CSCs, decrease proliferation, and promote apoptosis in metastatic tumors, and deplete CD44^+^CD24^−^ population. All these effects were enhanced when circRGPD6 was combined with docetaxel. circRGPD6 exerts its function by binding miR-26b and upregulation of YAF2. There was also an association found between clinicopathological features and circRGPD6 expression—non-metastatic breast cancer patients exhibited elevated circRGPD6 levels as compared to metastatic patients and those with high expression of circRGPD6 showed significantly longer disease-free survival and overall survival than those who demonstrated low expression. Use of circRGPD6 with docetaxel may serve as a new avenue in eradication of BCSCs and treatment of metastatic breast cancer [152].

#### 2.4.3. lncRNAs in CSCs

lncRNA, RNAs longer than 200 nucleotides which lack coding potential, have lately gained significant interest as potential regulators of various biological processes at both transcriptional and post-transcriptional level [153]. lncRNAs are a highly heterogenous class of RNAs not only due to a vast repertoire of possible functions, but also in the context of their biogenesis and origin [154,155]. These molecules can play both structural and regulatory roles by interaction with RNA (miRNA sponging, interaction with mRNA), DNA (formation of R-loops or triplexes), or with a variety of RNA- and DNA-binding proteins (RBPs and DBPs, respectively), affecting transcription of neighboring and distant genes, chromatin remodeling, and being involved in post-transcriptional regulation, and thus influencing splicing, mRNA stability, and translation and signaling pathways. Although lncRNA do not possess open reading frames (ORFs) their coding potential and possible implications in biological processes of the generated peptides have been recently reported [156,157,158].

As potent transcriptional and posttranscriptional regulators, lncRNAs are of key importance for CSCs biology. They control processes ranging from self-renewal and expression of stem cells’ characteristic markers, metabolism, and maintenance to chemoresistance, tumor-initiating capacity and epithelial-to-mesenchymal transition in which CSCs play a leading role as promoters or suppressors (Table 4).

##### lncRNAs as Promoters of CSCs

lncRNAs in breast cancer stem cells seems to be an intensive object of study in the field.

H19 is a hypoxia-related lncRNA which regulates stem cells features in breast cancer. Expression of this long non-coding RNA is upregulated in ALDH^+^ and SP breast cancer cells (5-10-fold as compared to ALDH^−^ and non-SP cells). Functional studies have revealed that H19 knockdown disrupts glycolysis as evident by reduction in cellular glucose uptake, lactate production, and ATP levels under hypoxia and downregulates breast cancer stemness as sphere-formation capacity was decreased and expression of *C-MYC, OCT4* and *LIN28* was downregulated under hypoxic conditions. In vivo, H19 enhances tumor growth and tumorigenic potential. Mechanistically, this long non-coding RNA acts as let-7 sponge to upregulate HIF-1α, and, subsequently, increase the levels of PDK1. As suggested by the authors, PDK1 is a key target in H19 mediated regulation of cancer stem cell maintenance and glycolysis. Furthermore, it has been demonstrated that by inhibition of H19 and PDK1, aspirin suppresses glycolysis and stemness of breast cancer. Clinically, H19 was upregulated in breast cancer tissues compared to normal tissues (5-fold) and its expression correlated with poor overall survival [159].

As to lncRNAs interfering with crucial stemness signaling pathways, by interaction with USP22 protein, LINC01426 influences Shh ubiquitination, and activates Hedgehog pathway in lung cancer stem cells. LINC01426 was found upregulated in both lung adenocarcinoma (LUAD) tissues and cell lines (5-fold as compared to normal tissue and >2-fold as compared to human bronchial epithelial cells). It promoted proliferation, migration, and inhibited apoptosis of LUAD cells, and influenced expression of EMT-associated markers. lncRNA facilitates self-renewal as evident by sphere formation form LUAD cells and increased levels of Nanog and Oct4 [161].

Long intergenic non-protein coding RNA 1106 (LINC01106) is implicated in colorectal cancer, and it is associated with Hedgehog pathway. LINC01106 is upregulated in colon adenocarcinoma (COAD) tissues (>1.5 fold as compared to normal/adjacent normal tissues) and negatively correlates with overall survival of COAD patients. LINC01106 silencing reduces the colony-forming ability, cell proliferation, and migration capacity and levels of migration-related proteins. Its knockdown possibly blocks EMT process as evident by the increase in E-cadherin levels and decrease in N-cadherin levels. As to CRC stemness, LINC01106 upregulates *NANOG* and *OCT4,* and facilitates sphere formation of CRC cells. Thus, lncRNA exerts its function via two main mechanisms—in the cytoplasm it positively regulates Gli4 by sequestering miR-449b-5p, and in the nucleus it recruits FUS to Gli1 and Gli2 promoters to activate transcription of these 2 Hedgehog downstream factors. Furthermore, it was revealed that Gli2 activated LINC01106 transcription [160].

##### lncRNAs as Suppressors of CSCs

Ma et al., 2019 has revealed that FGF13-AS1, downregulated in breast cancer tissue (>1-fold as compared to normal tissue), inhibits proliferation, invasive capacity, and metastasis of breast cancer cells in vitro and in vivo. FGF13-AS1 was shown to interfere with glycolysis, OCT4 and SOX2 expression, spheroid formation and maintenance of CD44^+^CD24^−^ population of breast cancer cells. Mechanistically, FGF13-AS1 interacts with IGF2BPs, m^6^A reader, to reduce Myc mRNA half-life. Myc protein also inhibits this lncRNA on the level of transcription. FGF13-AS1 expression was additionally found to correlate with lymph node metastasis and breast tumor stage and its low levels indicated worse outcomes in terms of relapse-free survival [162].

SCIRT (Stem Cell Inhibitory RNA Transcript) is a lncRNA which plays a role in transcriptional regulation of self-renewal and cell cycle-related genes important in transition between breast cancer stem cells and mature cancer cells characterized by higher proliferation rate. SCIRT has been shown to inhibit sphere formation in BC cells, reduce tumorigenesis in vivo and decrease migration. Knockdown of SCIRT leads to upregulation of genes involved in TGFβ and PIK3-Akt pathways, enriched in pluripotency factors and, the downregulation of genes involved in cell-cycle and DNA replication pathways (such as *FOXM1, E2F4,* or *E2F7*). SCIRT, through interaction with EZH2, recruits FOXM1 to induce transcription at cell-cycle-related genes’ promoters counteracting EZH2-mediated effects. Furthermore, SCIRT decreases transcription at self-renewal-related genes’ promoters antagonizing SOX2 and EZH2 effects by forming a complex with them [163].

## 3. Non-Coding RNAs in Therapeutic Anti-CSCs Strategies

According to the current understanding of CSCs, their crucial role in tumor development and metastasis, CSCs became one of the subjects pertaining to anti-cancer therapy. Presently, there are no effective treatments targeting CSCs. However, there are several clinical trials assessing drugs targeted to surface biomarkers, signaling pathways of CSCs, EMT process and microenvironment elements strongly connected to CSC biology [9]. Based on CSC surface markers, monoclonal antibodies directed against CD44 (SPL-108) and EpCAM (emovab) in ovarian cancer, or CD47 (TTI-621) and CD70 (ARGX-110) in acute myeloid leukemia are being evaluated. Wnt receptor and PI3K/mTOR, highly activated in cancer stem cells can be also promising targets [9]. For example, Ipafricept (OMP-54F28) is an inhibitor of Wnt receptor in clinical trial (NCT01608867). SAR245409 was tested in locally advanced or metastatic solid tumors as an inhibitor of PI3K/mTOR.

Use of non-coding RNAs is also a promising strategy to combat cancer stem cells. Non-coding RNAs exhibit specific expression and play roles in functioning of cancer stem cells, regulating their fundamental features, including resistance to conventional therapies, as can been seen from above presented research. This makes them attractive drug candidates and therapeutic targets.

There are two major approaches to the use of ncRNAs in cancer—generation of synthetic versions of tumor suppressive molecules to enhance their activity or targeting oncogenic RNA to silence their expression and, at the same time, to minimize their activity [164]. To ensure tumor-specific delivery several different strategies have been explored such as virus-based, lipid nanoparticles, nano-sized carriers, introduction of chemical modifications or bioconjugation to oligonucleotides [165,166].

miRNAs may be a potent therapeutic tool as one miRNA can regulate many mRNAs which translates into regulation of many cellular pathways. To modulate pathological levels of miRNAs, synthetic miRNAs (miRNA mimics) and oligonucleotide-based inhibitors of miRNAs (anti-miRs) are commonly used [167]. Furthermore, miRNA sponges and miRNA-masking antisense oligonucleotides seem promising tools. However, they have not yet reached clinical evaluation [168]. Several miRNA-based drugs with potential anti-tumor activities have been tested in clinical trials-MRX34, a miR-34 mimic; MesomiR 1, a miR-16 mimic; Cobomarsen/MRG-106, anti-miR-155, and INT-1B3, a miR-193a-3p mimic (Table 5).

siRNAs (short interfering RNAs) are also short single or double stranded RNAs, which silence expression of their targets via RISC (RNA-induced silencing complex). As opposed to miRNAs, they lead to degradation of complementary mRNAs only; therefore, they usually have few targets [169]. Currently, 17 clinical trials in different types of cancer are being conducted (phase I/II/III; status: completed/recruiting/active) (Table 6).

shRNAs (short hairpin RNA) utilize the siRNA/miRNA biogenesis pathway as these RNA hairpins are first processed into mature sequences by Dicer and then loaded into RISC [170]. 9 shRNAs are currently tested clinically in cancer (Table 6).

There are still several obstacles in the way to find a successful translation of RNA-based therapeutics into a clinic. One of the biggest challenges is specificity.

Candidate RNA therapeutics may cause off-target effects in a sequence-dependent and sequence-independent manner, and these include partial complementarity to undesired targets, and interactions with proteins, including immune defense-related. What may also occur is loading of the passenger strand instead of the guide strand into RISC, and toxicity, especially connected to cell proliferation [168,171].

Effective and safe delivery is another critical issue. By nature, oligonucleotides are hydrophilic and negatively charged which poses a problem for their transport via biological membranes. Additionally, they can be characterized by relatively short half-life. Another issue in this area, which is also connected to specificity are undesired on-target effects which result from systemic (as opposed to cell- or organ-specific) delivery of the therapeutic [168,172]. Here, it would be worth to mention a clinical trial of MRX34 (a miR-34a mimic) which was terminated due to severe immune-related side events and death of four patients. Possible explanation may lie in the fact that systemically delivered miR-34a mimic was taken up by white blood cells and it is now known that this miRNA impacts NF-KB pathway, a modulator of TCR-mediated signaling and cholesterol efflux in macrophages [173,174,175,176]. Nevertheless, cause of the effects observed in the study remains to be elucidated. There are now many new promising delivery strategies, including those which enable cell- and organ-specific delivery, and which are currently tested in clinical trials [172]. Some of them are not free from disadvantages, such as potential integration into the host genome in case of viral vectors, dose-limiting toxicities, or immune-related effects [177].

Potential immunogenicity is another feature of candidate therapeutic RNA molecules which may be a hurdle to their clinical application. As a viral defense mechanism, human immune system harbors intra- (TLRs 3, 7, 8, and 9 in endosomes) and extra-cellular (cytosolic sensors (e.g., protein kinase R, RIG-I, and MDA-5)) PAMP receptors which recognize single and double stranded RNAs. This interaction between the receptor and RNA causes immunological reactions, which are part of the natural defense, and which include release of cytokines, proliferation of immune cells, and activation of adaptive immunity [178]. Here, again, clinical trial of MRX34 can be provided as an example in which a miR-34a mimic as a short dsRNA potentially caused severe immune-related toxicities in patients [173,174].

Dosing is another important matter which should be taken into consideration. As many of RNA therapeutics make use of endogenous machinery responsible for RNAi, if high concentrations of such RNAs are to be administered, the machinery may become saturated, and this, in consequence, may lead to inhibition of endogenous small RNAs’ activity. In case of miRNAs, an additional issue arises due to their inherent feature of targeting multiple mRNAs and competition for binding of a particular target by multiple miRNAs. It remains unclear how changes in concentration of particular miRNA influence the whole array of its’ targets. Data on dose-dependent regulation of the targetome, described in a quantitative manner is still lacking [168,177].

It should be emphasized that many of those aspects have been partially addressed—new chemical modifications of RNA aiming at reduction in immunostimulation, increase in specificity to the target-of-interest, resistance to nuclease-mediated degradation, and increase in cellular uptake have been already incorporated [165]. As mentioned earlier, promising strategies which enable more effective and specific delivery have been developed and are now tested clinically, including liposomes, polymers, extracellular vesicles (EVs), nonliving minicells, viral vectors, conjugation of lipids, sugars, peptides, metal-based nanoparticles [172]. What remains critical is through testing of specificity of new potential therapeutics and possible expansion of screening of potential immunological effects [171].

Although challenges emerged at the beginning of the way to clinical application of RNA-based therapeutics, many of them have been already defined and begun to be addressed. What seems to be the key to success is an interdisciplinary approach which would utilize achievements of fields such as molecular biology, nanotechnology, immunology, or pharmacology.

circRNAs and lncRNAs have not yet reached clinical testing, they are however promising candidates for new therapies. Circularized RNA transcripts are already being generated and tested as potential therapeutics in vitro and in vivo, and these include synthetic circRNAs which can bind/sponge oncogenic miRNAs [152,179,180,181,182]. Several different approaches of their generation have been explored but inverted repeat-induced backsplicing seems to be chosen most [183]. Cancer-promoting circRNAs could be potentially silenced with RNAi, antisense technology, or CRISPR-Cas13 by targeting the unique sequence (backsplice junction) [183,184].

As to lncRNAs, they comprise two types of elements—interactor and structural—involved in interactions with RNA, DNA, and proteins to serve functions in, among others, translation, transcription of neighboring genes, and chromatin remodeling [154]. Such sequence motifs or secondary/tertiary structures may be short, and therefore could be crafted into RNA therapeutics [185]. Oncogenic lncRNAs, on the other hand, could be targeted by small molecules if secondary structures are taken into consideration or by antisense oligonucleotides (ASOs), in the context of sequence [168].

To conclude, cancer stem cells are characterized by many different features, like remarkable plasticity, metabolic changes, drug resistance, which are strongly linked with molecular alterations, ncluding complex non-coding RNAs influence. Understanding the biology of CSCs and mechanisms of their regulation by microRNAs, circRNAs or lncRNAs is crucial in terms of effective anticancer therapy. Hence, the number of promising therapeutic strategies targeting CSCs are currently under development.

## Figures and Tables

**Figure 1 cells-11-03699-f001:**
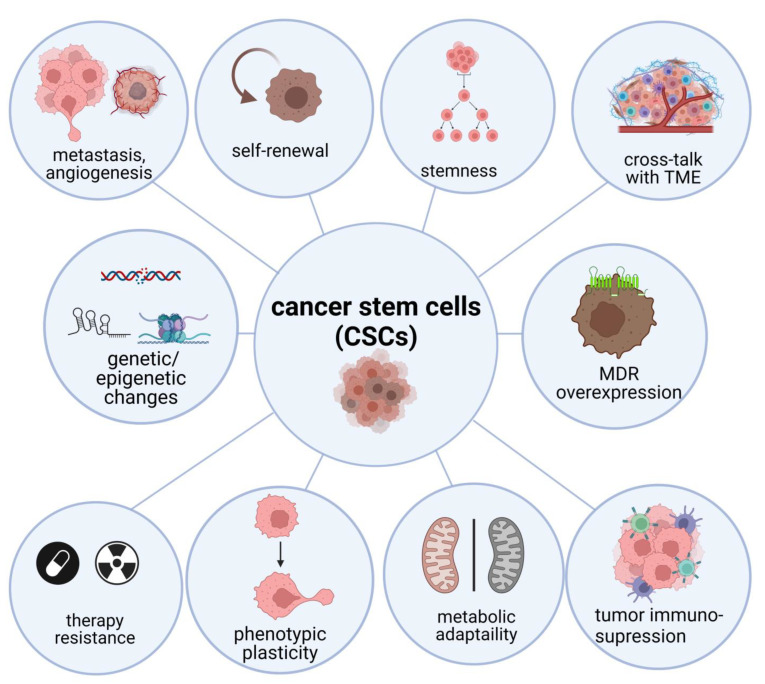
The features and mechanisms of CSCs biology. Abbreviations: MDR—multi-drug resistance proteins; TME—tumor microenvironment (Created with BioRender.com (accessed on 7 January 2022)).

**Figure 2 cells-11-03699-f002:**
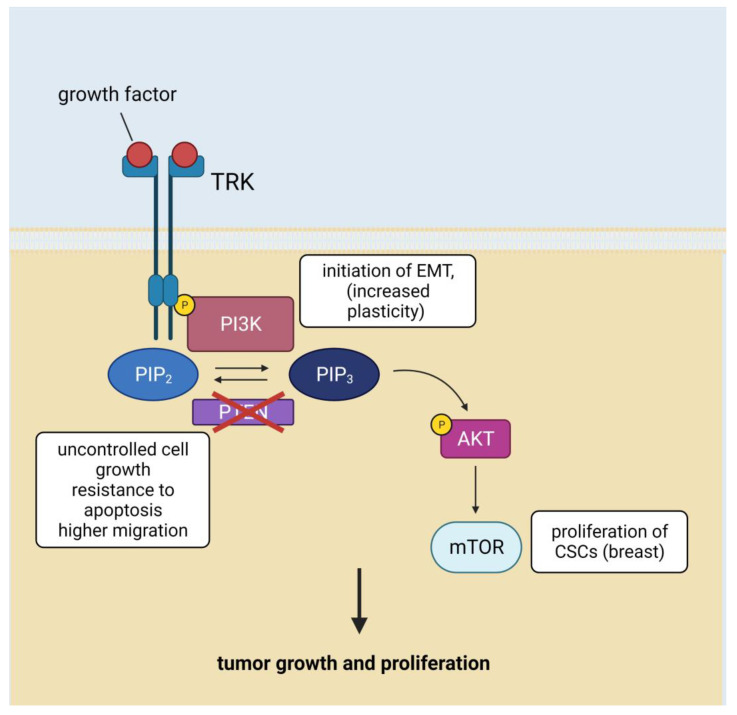
PI3K/Akt/mTOR signaling in cancer stem cells. Abbreviations: AKT—protein kinase B; EMT—epithelial-mesenchymal transition; mTOR—mammalian target of rapamycin; PI3K—phosphatidylinositol 3-kinase; PIP2—phosphatidylinositol 4,5-bisphosphate; PIP3—phosphatidylinositol (3,4,5)-trisphosphate; PTEN—phosphatase and tensin homolog; TRK—tyrosine kinase (Created with BioRender.com (accessed on 7 January 2022)).

**Figure 3 cells-11-03699-f003:**
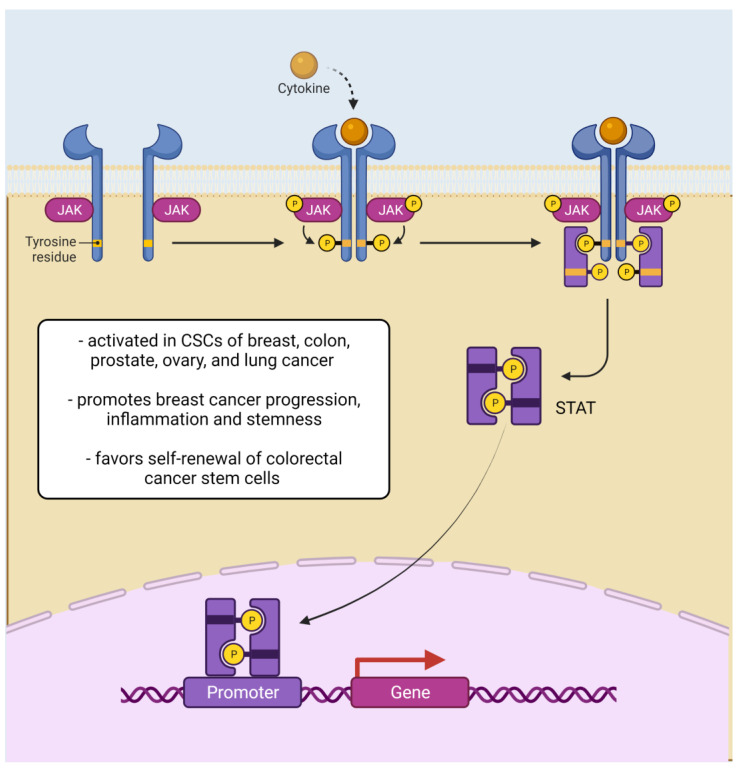
JAK-STAT signaling in cancer stem cells. Abbreviations: CSCs—cancer stem cells; JAK—Janus kinase; STAT—signal transducer and activator of transcription (Created with BioRender.com (accessed on 7 January 2022)).

**Figure 4 cells-11-03699-f004:**
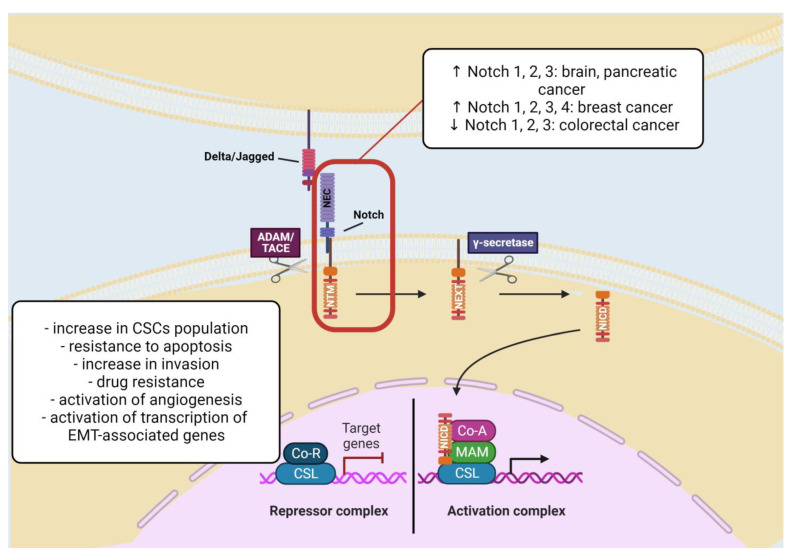
The Notch signaling pathway in cancer stem cells. Abbreviations: ADAM/TACE—a disintegrin and metalloproteinase/tumor necrosis factor-alpha converting enzyme; Co-A—co-activator; Co-R—co-repressor; CSCs—cancer stem cells; CSL—CBF1, Suppressor of Hairless, Lag-1; EMT—epithelial-to-mesenchymal transition; MAM—mastermind; NEC—Notch extracellular subunit.; NEXT—Notch extracellular truncation; NICD—Notch intracellular domain; NTM—Notch transmembrane subunit (Created with BioRender.com (accessed on 16 November 2022)).

**Figure 5 cells-11-03699-f005:**
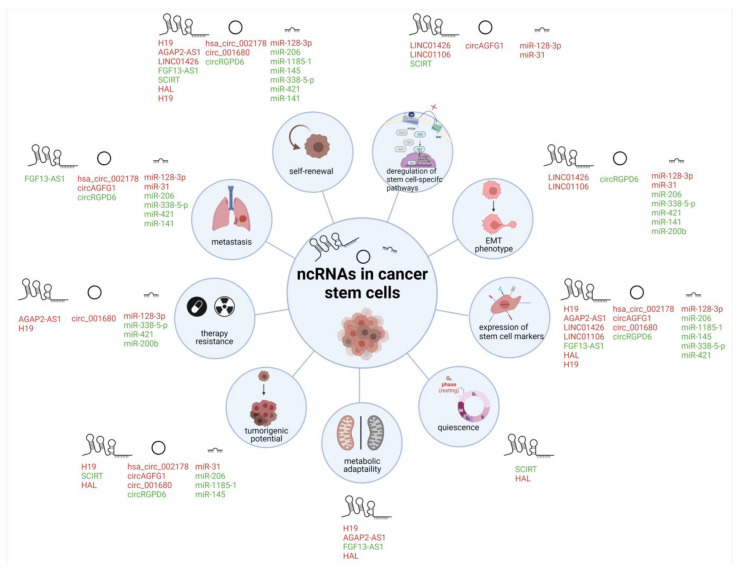
Impact of ncRNAs on cancer stem cells’ hallmarks. Abbreviations: EMT—epithelial-mesenchymal transition. ncRNAs with CSCs promoter role and with CSCs suppressor role are presented in red and green, respectively (Created with BioRender.com (accessed on 16 November 2022)).

**Table 1 cells-11-03699-t001:** Genes associated with CSCs plasticity.

Gene	Function	Reference
*SNAI1 (SNAIL)*	Promotes tumor growth, invasion, migration of cancer cells	[32]
*SNAI2 (SLUG)*	Prevents cell death and promotes cell survival	[33]
*ZEB1*	EMT transcription factor, causes plasticity of non-CSCs into CSCs, promotes stem-like and tumorigenic phenotype	[34]
*HIF1A*	Promotes the expression of stem cell-associated transcription factors, prompt the transcription of genes involved in the metabolism	[35,36]
*MCL1*	Promotes chemotherapy resistant CSC via the regulation of OXPHOS	[37]
*VEGF*	Invasion related to hypoxia	[38]
*MYC*	The driver of stemness and glycolysis	[39]
*PIK3CA*	Mutated PIK3CA induces reprogramming of lineage restricted progenitors to a multipotent stem-like state	[40]
*NOTCH*	Regulation of asymmetric division and cell stemness	[41]
*SOX2*	Maintaining pluripotency of stem cells	[26]
*SOX9*	Regulator of epithelial cell proliferation; acquiring properties of basal stem cells; induction of EMT.	[28]
*KLF4*	Promotes production of ECM that creates pro-metastatic niche	[42]
*MIF*	Conversion of cells phenotype from CD138- to CD138+	[43]
*STAT3*	Causes shift to aerobic glycolysis	[44]

**Table 2 cells-11-03699-t002:** Characteristics of chosen miRNAs in CSCs.

miRNA Name	Tumor Type	Target mRNAs	CSCs Promoter/Suppressor	Reference
miR-31	Breast	DKK1, AXIN1, GSK3Β	Promoter	[118]
miR-128-3p	Lung	AXIN1, SFRP2, WIF1, SMURF2, PP1c	Promoter	[119]
miR-145	Colorectal	SNAI1	Suppressor	[120]
miR-1185-1	Colorectal	CD24	Suppressor	[121]
miR-141	Prostate	CD44, EZH2, Rho GTPases (RAC1, CDC42, CDC42EP3 and ARPC5)	Suppressor	[122]
miRNA-338-5pmiRNA-421	Prostate	SPINK1	Suppressor	[123]
miRNA-34a, miRNA-34b/c	Colorectal	PDGFRA, PDGFRB, AXL, WASF1 (NGS-, reporter assay- and WB-validated) FGFR1, IGF1, STC1, CACNA2D2, COL6A2, COL4A2, INHBB (NGS- and qPCR-validated)	Suppressor	[124]

**Table 3 cells-11-03699-t003:** Characteristics of chosen circRNAs in CSCs.

circRNA Name	Tumor Type	Target	CSCs Promoter/Suppressor	Reference
hsa_circ_002178	Breast	miR-1258/KDM7A	Promoter	[149]
circAGFG1	Colorectal	miR-4262/miR-185-5p/YY1/CTNNB1; WNT/β-catenin pathway activation	Promoter	[150]
hsa_circ_001680	Colorectal	miR-340/BMI1	Promoter	[151]
circRGPD6	Breast	miR-26b/YAF2	Suppressor	[152]

**Table 4 cells-11-03699-t004:** Characteristics of chosen lncRNAs in CSCs.

lncRNA Name	Tumor Type	Mode of Action; Interacting with RNA/DNA/Protein	CSCs Promoter/Suppressor	Reference
H19	Breast	miRNA sponging-let-7/HIF-1α/PDK1	Promoter	[159]
LINC01106	Colorectal	Cytoplasmic: miRNA sponging-miR-449b-5p/Gli4; Nuclear: RBP binding-FUS/Gli1/Gli2	Promoter	[160]
LINC01426	Lung	Interacting with protein-USPP2/SHH/Hedgehog pathway activation	Promoter	[161]
FGF13-AS1	Breast	RBP binding-IGF2BPs/Myc	Suppressor	[162]
SCIRT	Breast	transcription induction- recruitment of FOXM1 through interaction with EZH2; transcription inhibition—interaction with SOX2 and EZH2 to antagonize their effects	Suppressor	[163]

**Table 5 cells-11-03699-t005:** Clinical trials of miRNA-based drugs in cancer.

Therapeutic Name	Mode of Action	Tumor Type	Delivery System	Clinical Trial Number	Phase, Recruitment Status
MRX34	miR-34 mimic	Melanoma; advanced and metastatic solid tumors	Intravenously/ vehicle transfer (liposomal)	NCT01829971 NCT02862145	Phase I T (serious adverse events)
MesomiR 1	miR-16 mimic	MPM and NSCLC	Intravenously/ vehicle transfer (nonliving minicells)	NCT02369198	Phase I C
Cobomarsen/ MRG-106	anti-miR-155	CTCL, CLL, DLBCL or ATLL	Intravenously/chemical modification (LNA)	NCT02580552 NCT03837457 NCT03713320	Phase I T (business reasons)
INT-1B3	miR-193a-3p mimic	Solid tumors	Intravenously/vehicle transfer (lipid NP)	NCT04675996	Phase I R

Abbreviations: ATLL—adult T-cell leukemia/lymphoma; CLL—chronic lymphocytic leukemia; CTCL—cutaneous T-cell lymphoma; DLBCL—diffuse large B-cell lymphoma; MPM—malignant pleural mesothelioma; NSCLC—non-small cell lung cancer. Clinical trial phase status: A-active, not recruiting; C-completed; R-recruiting; T-terminated.

**Table 6 cells-11-03699-t006:** Phase I completed/II/III clinical trials of shRNA and siRNA-based drugs in cancer—selected examples (data from clinicaltrials.gov (accessed on 28 August 2022)).

Therapeutic Name	Mode of Action	Tumor Type	Delivery System	Clinical Trial Number	Phase, Recruitment Status
Vigil ™ vaccine	Bi-shRNA-furin and GMCSF +carboplatin +bevacizumab +atezolizumab +irinotecan, temozolomide	Ewing’s sarcoma NSCLC Liver cancer Stage III/IV Ovarian cancer Ovarian, Cervical, Uterine cancer Ewing’s sarcoma	Intradermally/ (autologous tumor cells expressing Vigil plasmid)	NCT01061840 NCT01867086 NCT01551745 NCT03073525 NCT03495921	Phase I C Phase II C Phase II C Phase II A Phase III A
pbi-shRNA™ EWS-FLI1 Type 1 LPX	bi-shRNA EWS/FLI1	Ewing’s sarcoma	Intravenously/vehicle transfer (liposomal)	NCT02736565	Phase I A
lentivirus vector CCR5 shRNA-TRIM5alpha-TAR decoy	CCR5 shRNA	HIV infection, Lymphomas	Intravenously/transduced autologous CD34+ hematopoietic progenitor cells transplantation/vehicle transfer (lentivirus)	NCT02797470	Phase I/II R
lentivirus vector rHIV7-shI-TAR-CCR5RZ	shRNA targeted to an exon of the HIV-1 genes tat/rev	Lymphoma	Intravenously/transduced hematopoietic progenitor cells and non-bound CD34+ cells/vehicle transfer (lentivirus)	NCT00569985	Phase I C
pbi-shRNA ™ STMN1	STMN1 bi-shRNA	Advanced, metastatic Cancer, solid tumors	Intratumoral injection/transfer vehicle (bilamellar invaginated vesicle lipoplex (BIV LP))	NCT01505153	Phase I C
Atu027	PKN3 siRNA + gemcitabine	Advanced solid tumors	Intravenously/vehicle transfer (liposomes)	NCT00938574 NCT01808638	Phase I C Phase I/II C
ARO-HIF-2	TRiM (RGD- HIF-2α siRNA conjugate)	ccRCC	Intravenously/bioconjugation (alpha-v beta3 targeting ligand)	NCT04169711	Phase I C
siG12D-LODER	KRASG12D siRNA	Advanced pancreatic cancer	Intratumoral/Biodegradable polymeric matrix (PLGA)	NCT01188785 NCT01676259	Phase I C Phase II R
TKM 080301	PLK1 siRNA	Colorectal, pancreas, gastric, breast, ovarian cancer with hepatic metastase	Intravenously/vehicle transfer (LNP)	NCT01437007	Phase I C
APN401	Cbl-b siRNA	Inoperable metastatic solid tumors	Intravenously/siRNA-transfected Peripheral Blood Mononuclear Cells	NCT03087591 NCT02166255	Phase I C

Abbreviations: ccRCC—clear cell renal cell carcinoma; NSCLC—non-small cell lung cancer. Clinical trial phase status: A-active, not recruiting; C-completed; R-recruiting.

## Data Availability

Not applicable.

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
