# Peer review of "Cancer Stem Cells—The Insight into Non-Coding RNAs"

_cells, 2022, doi:10.3390/cells11223699_

Round 1

Reviewer 1 Report

This review is collection of information about CSC and regulation by non-coding RNAs. I have a few concerns.

1.      Because there are so many places, interpreting the meaning of sentences is tough. As a result, it is necessary to revise, clarify or need to add more information for better understanding, For example,

                                                    i.     Line no 107-112. Need to rewrite.

                                                   ii.     Line no 310-316. Need to rewrite.

                                                  iii.     Line no 81 to 84

Please check other places top

2.      Conclusion part need to re-write

3.      Why chapter 3 in line no 296. It is need to be clarify

4.      Sometimes, it is observed that the poor performance of non-coding RNAs during treatment, which need to be discussed in the review.

5.      How much particular non-coding RNA is effective in specific case of cancer that also need to be discussed, which is not should be in terms of up- or downregulation. It should in terms level or any clinical data.

6.      Authors must emphasise appropriate typing of each protein or gene's name. Genes are not italicised in several places and typographical errors in manuscript, which has to be formatted according to the journal.

7.      Table no 1. Authors can possibly write a gene or a protein, not both.

Reviewer 2 Report

In this manuscript, Bryl et al. collected information from the literatures about various genes/signaling/factors related to biology of CSCs, however, overall, the authors did not have enough deepness about the coverage of the 2 key topics, “plasticity and epigenetics”, which are in the title. The authors should give an accurate definition about the two concepts when they are first mentioned/described, and later link various genes/signaling/factors to the two terms, respectively. In this way, the manuscript will be more coherent with an emphasis on the two elements (plasticity and epigenetics of CSCs). Otherwise, the authors need to change the title to focus on more selected/narrowed topics.

Specific comments are as follows:

1) As plasticity of CSCs, phenotypic changes following alteration of tumor environment or therapy, is one the two important parts in the title of the manuscript, the authors should focus on this point when they discuss various genes/signaling/factors that are related to CSCs.  In addition, it is recently believed that plasticity of CSCs may include dormancy and reactivation of CSCs following therapy and could represent the cause/reason of therapeutic resistance and caner recurrence.  The latter could be regarded as an improvement of original CSC theory. The authors are encouraged to elaborate more on plasticity, dormancy and reactivation of CSCs following therapy or environmental changes. There are intracellular (internal) and extracellular (external) factors influencing/determining the plasticity of CSCs.

2) Line 63, the subtitle “Biology of CSCs and their influence on tumor microenvironment” is not an appropriate one.  Plasticity or biology of CSCs is regulated/influenced by tumor environment, although they may interact with the latter.

3) Line 72, the description “Cancer stem cells’ phenotypic plasticity is a driving force of cancer initiation” is not correct. driving force means  molecules/substances but not plasticity/features. Plasticity is a reason/cause/mechanism, but not a driving force.

4) EMT is a typical feature and mechanism of CSC plasticity, the authors are encouraged to elaborate more on this part.

5) When epigenetics is discussed, the authors should give a good description about epigenetics. It seems that the authors missed other important parts of epigenetics, for example, histone methylation and related enzymes and etc. Otherwise, the authors need to narrow the concept/topic in the title.

6) Line 210, the subtitle, “2.4. Epigenetic changes in CSCs – the role of microRNAs, circRNAs and lncRNAs” is not appropriate, should be changed to “2.4. Epigenetic regulation of CSCs – the role of microRNAs, circRNAs and lncRNAs”

7) The following papers can be considered to be included:

A)“EMT related to stem cells or CSCs”

Sun Y et al. (2012) Androgen deprivation causes epithelial-mesenchymal transition in the prostate: implications for androgen-deprivation therapy. Cancer Res. Jan 15;72(2):527-36. 

B) Guo Y et al. (2017) Numb-/low enriches a castration resistant prostate cancer cell subpopulation associated with enhanced Notch and Hedgehog signaling. Clin Cancer Res.. 23:6744-6756.  

“Numb is an inhibitor of notch signaling, related to Notch and CSCs”.

Round 2

Reviewer 1 Report

I am satisfied with the author's response. I just have one suggestion: enhance the text size in Figures 1 and 5 for better readability.

Reviewer 2 Report

The authors have essentially addressed my concern and the quality of the manuscript has been improved.